


# Contrasting effects of CO$_2$ fertilisation, land-use change and warming on seasonal amplitude of northern hemisphere CO$_2$ exchange

Ana Bastos[1], Philippe Ciais[2], Frédéric Chevallier[2], Christian Rödenbeck[3], Ashley P. Ballantyne[4], Fabienne Maignan[2], Yi Yin[5], Marcos Fernandez-Martinez[6], Pierre Friedlingstein[7], Josep Peñuelas[8,9], Shilong L. Piao[10], Stephen Sitch[11], William K. Smith[12], Xuhui Wang[2], Zaichun Zhu[13], Vanessa Haverd[14], Etsushi Kato[15], Atul K. Jain[16], Sebastian Lienert[17], Danica Lombardozzi[18], Julia E. M. S. Nabel[19], Philippe Peylin[2], Ben Poulter[20], and Dan Zhu[2]

[1]Ludwig-Maximilians Universität, Deptartment of Geography, Luisenstr. 37, 80333, München, Germany
[2]Laboratoire des Sciences du Climat et de l'Environnement (LSCE), CEA-CNRS-UVSQ, UMR8212, 91191 Gif-sur-Yvette, France
[3]Max Planck Institute for Biogeochemistry, Jena, Germany
[4]Department of Ecosystem and Conservation Science, University of Montana, Missoula, Montana, 59812, USA
[5]Department of Environmental Science and Engineering, California Institute of Technology, Pasadena, CA 91125
[6]Centre of Excellence PLECO (Plants and Ecosystems), Department of Biology, University of Antwerp, 2610 Wilrijk, Belgium.
[7]College of Engineering, Mathematics and Physical Sciences, University of Exeter, Exeter EX4 4QF, UK
[8]CSIC, Global Ecology Unit CREAF-CEAB-UAB, Cerdanyola del Valles, 08193, Catalonia, Spain
[9]CREAF, Cerdanyola del Valles, 08193, Catalonia, Spain
[10]Sino-French Institute for Earth System Science, College of Urban and Environmental Sciences, Peking University, Beijing, China
[11]College of Life and Environmental Sciences, University of Exeter, Exeter EX4 4RJ, UK
[12]School of Natural Resources and the Environment, University of Arizona, Tucson, AZ 85721, USA
[13]School of Urban Planning and Design Shenzhen Graduate School, Peking University Shenzhen, 518055, P. R. China
[14]CSIRO Oceans and Atmosphere, Canberra, 2601, Australia
[15]Institute of Applied Energy (IAE), Minato, Tokyo 105-0003, Japan
[16]Department of Atmospheric Sciences, University of Illinois, Urbana, IL 61801, USA
[17]Climate and Environmental Physics, Physics Institute and Oeschger Centre for Climate Change Research, University of Bern, Bern CH-3012, Switzerland
[18]Climate and Global Dynamics Division, National Center for Atmospheric Research, Boulder, CO 80302, USA
[19]Max Planck Institute for Meteorology, 20146 Hamburg, Germany
[20]NASA Goddard Space Flight Center, Biospheric Sciences Lab., Greenbelt, MD 20816, USA

**Correspondence:** Ana Bastos (ana.bastos@lmu.de)

**Abstract.** Continuous atmospheric CO$_2$ monitoring data indicate an increase in seasonal-cycle amplitude (SCA) of CO$_2$ exchange in northern high latitudes. The major drivers of enhanced SCA remain unclear and intensely debated with land-use change, CO$_2$ fertilization and warming identified as likely contributors. We integrated CO$_2$-flux data from two atmospheric inversions (consistent with atmospheric records) and from and 11 state-of-the-art land-surface models (LSMs) to evaluate
5   the relative importance of individual contributors to trends and drivers of the SCA of CO$_2$-fluxes for 1980-2015. The LSMs generally reproduce the latitudinal increase in SCA trends within the inversions range. Inversions and LSMs attribute SCA



increase to boreal Asia and Europe due to enhanced vegetation productivity (in LSMs) and point to contrasting effects of $CO_2$ fertilisation (positive) and warming (negative) on SCA. Our results do not support land-use change as a key contributor to the increase in SCA. The sensitivity of simulated microbial respiration to temperature in LSMs explained biases in SCA trends, which suggests SCA could help to constrain model turnover times.

## 1   Introduction

The increase in the amplitude of seasonal atmospheric $CO_2$ concentrations at northern high latitudes is one of the most intriguing patterns of change in the global carbon (C) cycle. The seasonal-cycle amplitude (SCA) of atmospheric $CO_2$ in the lower troposphere at the high-latitude monitoring site of Point Barrow, Alaska, has increased by about 50% since the 1960s (Keeling et al., 1996; Dargaville et al., 2002). Increasing SCA has also been registered at other high-latitude sites, mostly above 50°N (Piao et al., 2017) and appears to be driven primarily by changes in seasonal growth dynamics of terrestrial ecosystems (i.e., net biome productivity, NBP), but uncertainty remains about the relative contributions from different continents, and mechanisms.

Some studies proposed that the trend in SCA is primarily driven by increased natural vegetation growth and forest expansion at high-latitudes due to $CO_2$ fertilization and climate change (Graven et al., 2013; Forkel et al., 2016; Piao et al., 2017). Others (Gray et al., 2014; Zeng et al., 2014) suggested that agricultural expansion and intensification resulted in increased productivity and thus enhanced the seasonal exchange in cultivated areas at mid-latitudes. However, evidence suggests that crop productivity stagnated after the 1980s in many regions in the Northern Hemisphere (Grassini et al., 2013), which is not reflected in SCA trends in recent decades (Yin et al., 2018).

Studies using land-surface models (LSMs) to attribute trends to the suggested processes usually convert simulated fluxes to $CO_2$ concentrations using atmospheric transport models (ATM) and compare the results to in-situ measurements (Dargaville et al., 2002; Forkel et al., 2016; Piao et al., 2017) or over latitudinal transects (Graven et al., 2013; Thomas et al., 2016). These studies have shown that LSMs systematically underestimated SCA trends, but it is not clear whether these biases are due to LSM uncertainties or due to trends or errors in the ATM (Dargaville et al., 2002). Piao et al. (2017) addressed these problems by designing systematic model experiments to compare observed $CO_2$ concentrations at multiple sites with ATM simulations forced by an ensemble of NBP from different LSMs and an ocean biogeochemistry model. Point Barrow was the only site where nearly all models accurately described the trend in SCA, while in other sites, LSMs generally captured the sign of the trend in SCA but either under- or over-estimated its magnitude. Piao et al. (2017) further reported that $CO_2$ fertilisation and climate change drove the increase in SCA for sites >50°N, but that at mid-latitude sites land use, oceanic fluxes, fossil-fuel emissions, as well as trends in atmospheric transport may have contributed to the SCA trends.

Atmospheric inversions provide a consistent framework for assimilating in-situ $CO_2$ concentration observations to estimate net $CO_2$ surface fluxes while accounting for errors in the prior fluxes and for some errors in the ATM (Peylin et al., 2013). SCA trends have a strong latitudinal gradient that is interpreted as spatio-temporal trends of surface flux seasonality by atmospheric inversions. The spatiotemporal distribution of terrestrial and oceanic surface fluxes estimated by inversions provides thus direct insight about the regional patterns of trends in the seasonal amplitude of $CO_2$ fluxes (i.e. SCA of NBP, $SCA_{NBP}$) that



is fully consistent with the amplitude of $CO_2$ concentrations in all stations of the observational network used and constitute a direct benchmark for $SCA_{NBP}$ simulated by LSMs. Here, we use top-down (inversions (Chevallier et al., 2010; Rödenbeck, 2005)) and bottom-up (TRENDYv6 LSMs (Le Quéré et al., 2018)) estimates of terrestrial $CO_2$ fluxes at northern extra-tropical latitudes between 1980-2015 to:

(i) assess the ability of those LSMs to simulate inversion-based trends in $SCA_{NBP}$;

(ii) attribute the trends in $SCA_{NBP}$ to specific regions in the Northern Hemisphere;

(iii) attribute the relative importance of drivers using the ensemble model framework.

The two inversions used here – the Copernicus Atmosphere Monitoring Service (CAMS) inversion system (Chevallier et al., 2005) and the Jena CarboScope inversion (Rödenbeck, 2005) – solve for fluxes on their ATM grid, thus minimising aggregation

errors for large regions (Kaminski and Heimann, 2001). The two inversions differ in a number of characteristics (see Methods), particularly the ATM, the prior information and the observations assimilated: CAMS includes a time-varying number of multi-year air-sampling sites as they are available (constraining better spatial patterns), and CarboScope keeps a fixed set of sites covering a given period (avoiding artefacts in the time series related to the appearance or disappearance of measurement sites).

We use NBP simulated by a set of 11 LSMs from the recent TRENDYv6 inter-comparison (Le Quéré et al., 2018) in

three distinct experiments where models are forced with: changing $CO_2$ only (S1); $CO_2$ and changing climate (S2); changing climate, $CO_2$, and LULCC (S3). Some of the TRENDYv6 models used here simulate some key management processes (4 models include wood harvest, 1 model irrigation, and 2 fertilization) and more models now include nitrogen cycling, and improved soil processes, which were missing in previous intercomparisons (Arneth et al., 2017) (see Sec. 2.2).

## 2   Data

### 2.1   Atmospheric inversions

The inversion of a transport model to infer surface fluxes from concentration measurements is an ill-posed problem due to the dispersive nature of transport in the atmosphere and to the finite number of available measurements. This *ill-posedness* can be compensated by using some prior information about the fluxes to be inferred. This prior information also drives the separation between natural and fossil fuel emissions in the estimation. In order to illustrate the diversity of the inversion results, we

take the example of two inversions systems that provide results for the study period between 1980 and 2015. We analysed monthly surface $CO_2$ fluxes estimated by the inversion systems from the Copernicus Atmosphere Monitoring Service (CAMS) (Chevallier et al., 2005, 2010) and from Jena CarboScope (Rödenbeck et al., 2003; Rödenbeck, 2005).

Here we use CAMS version r16v1 (http://atmosphere.copernicus.eu/) selected for the period between 1980-2015, that provides estimates of ocean and terrestrial fluxes at 1.9° latitude ×3.75° longitude resolution. The CAMS inversion system as-

similates observations from a variable number of atmospheric $CO_2$ monitoring sites (119 in total providing at least 5 years of measurements) and uses the transport model from the LMDz General Circulation Model (LMDz5A) nudged to ECMWF-analysed winds. More details can be found in (Chevallier et al., 2010).



The CarboScope v4.1 (available at http://www.bgc-jena.mpg.de/CarboScope/?ID=s) provides several versions that assimilate a temporally consistent set of observations. We used these versions for the study period (1980-2015) to test the influence of the number of assimilated sites on the results. The s76, s85, and s93 versions have assimilated observations from 10, 23, and 38 sites since 1976, 1985, and 1993, respectively. Surface fluxes (ocean and land) are provided at the latitude/longitude resolution of 4°×5° of the TM3 atmospheric transport model is used (Rödenbeck, 2005). In this version, the atmospheric model is forced by the National Centers for Environmental Prediction (NCEP) meteorological fields.

CarboScope further provides a sensitivity analysis of the s85 version fluxes to different parameters of the inversion. The sensitivity tests performed are: "oc" – fixing the ocean prior; "eraI" - forcing the inversion with fields from ERA-Interim reanalysis instead of NCEP; "loose" and "tight" - scaling the a-priori sigma for the non-seasonal land and ocean flux components by 4 (dampening) and 0.25 (amplification), respectively; "fast" - reducing the length of a-priori temporal correlations; "short" - reducing the length of a-priori spatial correlations. The resulting latitudinally-integrated $SCA_{NBP}$ and respective trends are shown in Figure S2.

## 2.2 Land-surface Models

Land-surface models (LSMs) provide a bottom-up approach to evaluate terrestrial $CO_2$ fluxes (i.e. net biome productivity, NBP), and allow deeper insight into the mechanisms driving changes in C-stocks and fluxes. The TRENDY intercomparison project compiles simulations from state-of-the-art LSMs to evaluate terrestrial energy, water and $CO_2$ exchanges since the pre-industrial period (Sitch et al., 2015; Le Quéré et al., 2018). We use LSMs from the TRENDY v6 simulations for 1860-2015. To identify the contributions of $CO_2$ fertilisation, climate, and LULCC and management to the observed changes in $SCA_{NBP}$, we use outputs from three factorial simulations.

The models in simulation S3 were forced by (i) atmospheric $CO_2$ concentrations from ice core data and observations, (ii) historical climate reanalysis from the CRU-NCEP v8 (Viovy, 2016; Harris et al., 2014) and (ii) human-induced land-cover changes and management from a recent update of the Land-Use Harmonization (Hurtt et al., 2011) prepared for the next set of historical CMIP6 simulations, LUH2v2h (described below). Most models still do not represent many of the management processes included in LUH2v2h, though. As summarized in Table A1 in (Le Quéré et al., 2018), four models do not simulate wood-harvest, and three do not simulate cropland harvest. Two models simulate crop fertilization, tillage and grazing.

The models in simulation S2 were forced by (i) and (ii) with fixed land-cover map from 1860. Simulation S2 estimates "natural" fluxes, and the difference between S2 and S3 outputs corresponds to anthropogenic $CO_2$ fluxes from LULCC. The models in simulation S1 were forced by changing atmospheric $CO_2$ and no climate change (recycling 1901-1920 values to simulate interannual variability) or LULCC. S1 thus provided changes in the terrestrial sink due to $CO_2$ fertilisation, and the difference between S1 and S2 indicates the influence of climate change only. However, management practices (e.g. wood-harvest), when simulated, are already included in S1 and S2 for some models. A baseline simulation with none of these effects (S0) was also performed to check for residual variability and trends. We selected only models providing spatially-explicit outputs for the four simulations (S0, S1, S2 and S3) at monthly intervals (to evaluate seasonality, Supplementary Table 1).





We used NBP outputs selected for the period common to the inversion data, i.e. 1980-2015. NBP corresponds to the simulated net atmosphere-land flux (positive sign for a $CO_2$ sink), i.e. gross primary productivity (GPP) minus total ecosystem respiration (TER), fire emissions and fluxes from LULCC and management (e.g. deforestation, agricultural and wood harvest, and shifting cultivation). All model outputs were resampled to a common regular latitude/longitude grid of 1×1°.

## 2.3   Land cover and management

### 2.3.1   LUH2v2h

The LUH2v2h (Hurtt et al., 2011) (available at http://luh.umd.edu/) provides historical states and transitions of land use and management in a regular latitude/longitude grid of 0.25×0.25°, covering 850-2015 at annual time intervals. Land-use states distinguish between primary and secondary natural vegetation (and forest and non-forest sub-types), managed pastures and rangelands, and multiple crop functional types. The updated data set includes several new layers of agricultural management, such as irrigation, nitrogen fertilisation, and biofuel management, and spatially explicit information about wood harvest constrained by LANDSAT data. Each LSM, however, may not simulate all the processes introduced in LUH2v2h, so the S3 results from each simulation might not be directly comparable.

### 2.3.2   ESA-CCI Land-Cover

Land-cover information in LUH2v2h is combined with partial information on land use (e.g. rangeland in LUH2v2h can be either grassland or shrubland with low grazing disturbance). We therefore compared this information to annual land-cover maps at a latitude/longitude resolution of 0.5×0.5° based on the 300-m satellite-based land-cover data sets from ESA-CCI LC (https://www.esa-landcover-cci.org/?q=node/175) for 1992-2015. Data are provided for different vegetation types, but here were aggregated for four main land-cover classes: forest, shrubland, grassland, and cropland. The average distribution of these classes (forest and shrubland aggregated for readability) is shown in Figure 2a. LUH2 was used for the statistical analysis of inversion and the LSM drivers (because it was the data set used to force the models), and ESA-CCI data were used for the analysis of satellite-based vegetation data sets.

## 2.4   Satellite-based vegetation datasets

We further evaluated trends in the activity and growth of vegetation for the different land-cover classes using three satellite-based data sets: leaf-area index (LAI), net primary production (NPP), and aboveground biomass (AGB) stocks. The LAI data set was calculated from satellite imagery from Global Inventory Modeling and Mapping Studies (GIMMS LAI3g) described by (Zhu et al., 2015) for 1982-2015. LAI data were provided in two time-steps per month on a regular latitude/longitude grid of 1/12° (subsequently aggregated to 0.5°). (Smith et al., 2016) used the MODIS NPP algorithm and data for LAI and the fraction of photosynthetically active radiation from GIMMS to produce a 30-year global NPP data set, provided at monthly timescales for 1982-2011 at a latitude/longitude resolution of 1×1°. The data are available at the NTSG data portal (https://wkolby.org/data-code/). AGB stocks can be derived from estimates of vegetation optical depth derived from passive-





microwave satellite measurements. (Liu et al., 2015) produced a 20-year data set of AGB stocks for 1993-2012 based on measurements from a series of passive-microwave sensors. The data set is provided at a latitude/longitude resolution of 0.25×0.25° in annual time intervals and is available at http://www.wenfo.org/wald/global-biomass/ (last access 13/02/2018). We tracked changes in LAI, NPP, and AGB stocks for different land-cover types over time by selecting periods of at least 20 years common to ESA-CCI LC and the vegetation data sets (1992-2012 for LAI, 1992-2011 for NPP, and 1993-2012 for AGB stocks). Vegetation variables were then aggregated for the four land-cover types at each time interval to account for land-cover changes.

## 3   Methods

### 3.1   Trends in seasonal-cycle amplitude (SCA)

The seasonal amplitude of $CO_2$ concentration is modulated by higher ecosystem $CO_2$ uptake during the growing season and increased emissions during the release period (TER) and thus controlled by the seasonal amplitude of NBP. We calculated $SCA_{NBP}$ as the difference between peak uptake and trough for each year, at pixel scale shown in Figure 1a. However, since inversion fluxes have large uncertainty at pixel-level we focused our analysis on SCA trends estimated from aggregated NBP over latitudinal bands or Transcom3 regions (Baker et al., 2006). Because we do not impose the timing of peak and trough, changes in $SCA_{NBP}$ can be affected by the relative phase changes of GPP versus TER.

The trend in $SCA_{NBP}$ was calculated by a least-squares linear fit of annual values for 1980-2015, and confidence intervals were calculated based on the Student's t-distribution. We tested the robustness of estimated trends of inversions and LSMs for shorter periods by removing the first and last 1-10 years and trends of interannual variability by randomly removing 5 and 10 years of data 104 times. The significance of these trends was calculated using a Mann-Kendall test. We also compared different versions of CarboScope to evaluate the influence of the assimilated network size on the $SCA_{NBP}$ trends (Figure S1). We further calculated the trends for each of the sensitivity tests from CarboScope s85.

### 3.2   Process attribution

The three TRENDY experiments allow evaluating separately the effects of $CO_2$ fertilisation, climate change, and LULCC in the models. The differences between S1 and S2 and between S2 and S3, however, could not isolate specific processes that may have contributed to the trend (e.g. cropland expansion versus afforestation, or precipitation versus temperature). Furthermore, the LSMs may miss or simulate poorly certain processes that could influence $SCA_{NBP}$. Therefore, the attribution of drivers by the models is uncertain and should be cross-evaluated. Because inversions do not allow such partitioning between processes, a possible solution is to compare statistical attribution to drivers in inversions and LSMs.

We therefore compared the sensitivity of $SCA_{NBP}$ estimated by the inversions and the LSMs by fitting a general linear model (GLM) using the iteratively reweighted least squares method to eliminate the influence of outliers (Gill, 2000; Green, 1984). We tested the following variables (after unity-based normalisation) as predictors: fertilization, irrigation, wood harvest, growing-season precipitation, growing-season temperature, atmospheric $CO_2$ concentration, change in extent of cropland and





forest. These variables were taken from the corresponding datasets used to force TRENDYv6 models. All possible combinations of n predictors (n=1,2, ..., 7) were tested, and for each value of n, the "best" model (according to Akaike's information criterion) was chosen separately for each dataset. Above n=4 no model showed improved fit compared to the models with less predictors. The coefficients from the GLM fit for each dataset are shown in Figure S.

We further tested the robustness of the statistical relationships by fitting the GLM to the differences between each TRENDYv6 experiment. The significant predictors in the GLM fit to the LSMs in S3 should be detected in the corresponding factorial simulations, e.g. predictors associated with climate should be consistent for the fluxes estimated by the difference between S2 and S1 (effects of climate). The GLM fit to the partial fluxes for the effects of LULCC (S3-S2), climate (S2-S1), and $CO_2$ fertilisation (S1-S0) are shown in Figure S5.

## 4    Results and discussion

### 4.1    Large scale patterns

#### 4.1.1    Top-down estimates

Both inversions estimate increasingly positive trends in $SCA_{NBP}$ with increasing latitude, even though CAMS shows heterogeneous patterns in North America (Figure 1a, S1), and strong decreasing trends for mid-latitudes. Both inversions agree

on significant positive $SCA_{NBP}$ trends north of 40°N (defined here as band $L_{>40N}$) and non-significant trends for 25-40°N (band $L_{25-40N}$, Figures 1b). In the $L_{>40N}$ band, CAMS and CarboScope s76 v4.1 estimate an $SCA_{NBP}$ increase of 17.3±4.5 TgC.yr$^{-2}$ and 13.3±3.3 TgC.yr$^{-2}$, respectively. The uncertainties given for $SCA_{NBP}$ trends represent here the uncertainty of the linear fit due to the year-to-year $SCA_{NBP}$ variability (Methods). The difference between the CAMS and CarboScope inversions reflects part of the uncertainty in inversions due to their different choices in the ATM (including different atmospheric

forcing and spatial resolution), the set of assimilated $CO_2$ data, the prior fluxes, and the a-priori spatial and temporal correlation scales, and is comparable to the uncertainty of the linear fit due to inter-annual variability.

This finding is corroborated by two further analyses of inversion uncertainties:

(1) While both inversions assimilate atmospheric $CO_2$ measurements from Point Barrow, CAMS increasingly assimilates many other sites in the NH as they become available, helping to better constrain the $CO_2$ fluxes in mid- to high-latitudes with

time. Assimilating a non-stationary network of stations, however, possibly leads to spurious additional trends in $SCA_{NBP}$. To test this, we use different runs provided by CarboScope using more sites (but still fixed in number for each run, Figure S2) for more recent periods. The results from CarboScope version s85 v4.1 (1985-2015) are generally consistent with CAMS, but version s93 v4.1 (1993-2015) estimates much stronger $SCA_{NBP}$ trends (Table S1). A higher $SCA_{NBP}$ trend in the period 1993-2015 is reported by both CAMS and CarboScope, which estimate very similar trends in $L_{>40N}$ (19.5 TgC.yr$^{-2}$ and 19.2

TgC.yr$^{-2}$ respectively).

(2) CarboScope provides a set of sensitivity runs for s85 v4.1, varying some of the inversion's parameters (Figure S3). Changes in the meteorological fields driving the transport model and the prior ocean fluxes have the largest effect on the



$SCA_{NBP}$ trends, giving $L_{>40N}$ trends of 8.6±4.9 TgC.yr$^{-2}$ (ERA-Interim instead of NCEP) and 13.9±5.6 TgC.yr$^{-2}$ (fixed ocean), respectively, both well within the uncertainty range (interannual variability affecting linear fit to $SCA_{NBP}$ trend) estimated by the standard CarboScope s85 v4.1 (11.7±5.0TgC.yr$^{-2}$).

In summary, the ability of inversions to quantify the $SCA_{NBP}$ trend is mostly limited by the intrinsic year-to-year $SCA_{NBP}$ variability, less so by the amount of information available through the atmospheric data or by inversion settings.

### 4.1.2 Bottom-up estimates

The large-scale patterns of $SCA_{NBP}$ trends from the LSM Multi-Model Ensemble Mean (MMEM) of simulation S3 (all forcings) are consistent with inversions, especially with CarboScope (Figure 1a). The MMEM estimates are within the range of the inversions for most latitudes (Figure S1), but always at the lower end of $SCA_{NBP}$ trends reported by inversions. Consistent with inversions, LSMs report a significant trend in $L_{>40N}$ and a very weak (non-significant) trend in $SCA_{NBP}$ in $L_{25-40N}$ (Figure 1b). The overall MMEM trend in $L_{>40N}$ is significantly lower than in inversions (9.5±3.4 TgC.yr$^{-2}$, i.e. 55-71% of inversions' estimates). The agreement between LSMs and inversions also varies depending on the period and set of inversions considered (LSMs capture 65-91% of inversion trends in 1985-2015 and 74-75% in 1993-2015, Table 1). The MMEM estimate for 1985-2015 (10.6±4.5 TgC.yr$^{-2}$) is in fact, even higher than the CarboScope inversion with different meteorological fields (8.6±4.9 TgC.yr$^{-2}$). These results indicate that, despite a general underestimation of $SCA_{NBP}$ trend in $L_{>40N}$ during 1980-2015 as compared to top-down estimates, the LSMs simulate the main spatiotemporal patterns in $SCA_{NBP}$ trends consistent with inversions estimates, especially when accounting for the uncertainty in the latter.

To understand if recent improvements to the set of LSMs and their forcing in TRENDYv6 may have improved their performance in reproducing the $SCA_{NBP}$ trend, we compared $SCA_{NBP}$ trends from the previous intercomparison round (TRENDYv4). The MMEM from v6 estimates an $SCA_{NBP}$ trend in $L_{>40N}$ 43% higher than in than v4 (MMEM shown in Table S1, but evaluated for individual models). The specific reasons for improvement are hard to identify because of multiple model-dependent changes in the forcing, process simulation and parameterizations from v4 to v6 (Table 4 in (Le Quéré et al., 2018)).

In summary, we showed that the TRENDYv6 ensemble mean $SCA_{NBP}$ trend captures the positive trends in the high latitudes and the lack of trend in the mid-latitudes given by inversions, and under-estimates the magnitude of the high latitudes $SCA_{NBP}$ trends by 9-45%, depending on the inversion considered and period analysed.

### 4.2 Regional attribution

The comparison of $SCA_{NBP}$ trends in large latitudinal bands may be useful in diagnosing general patterns, but is less useful to diagnose drivers of trends (e.g. climate, agriculture), since ecosystem composition, land management and climate effects are not necessarily separated along a latitudinal gradient. However, the comparison of inversions and models at pixel scale is also not advisable, because the sparse atmospheric network does not allow constraining the fluxes at this scale. We thus compared inversions and LSMs for the $SCA_{NBP}$ trends over five sub-continental scale regions: boreal and temperate Eurasia and North



America regions, and Europe ("TransCom3" regions, Figure 2). We then use LSMs for attributing $SCA_{NBP}$ trends to different drivers using their factorial simulations (Methods).

Inversions and LSMs consistently attribute the increase in $SCA_{NBP}$ mainly to boreal Eurasia, both in area specific (Figure 1a) and integrated values (Figure 2b, 5.3-7.1 TgC.yr$^{-2}$ for inversions and 4.6 TgC.yr$^{-2}$ for MMEM, respectively) and to Europe

(1.9-3.7 TgC.yr$^{-2}$ and 2.3 TgC.yr$^{-2}$). The LSMs ascribed the trends in boreal Eurasia approximately equally to climate change and $CO_2$ fertilisation (S1 and S2), with LULCC having a slight negative (i.e. decreasing) effect (compare S2 and S3), consistent with the results by (Piao et al., 2017). In Europe, LSMs indicate negative contributions from both climate and LULCC. The negative effect of climate may be linked to increasingly drier conditions in this region (Greve et al., 2014) and to strong heatwaves in Europe in the early 21$_{th}$ century (Seneviratne et al., 2012). The negative contribution of LULCC indicated by

LSMs in Europe does not support the idea that agricultural intensification or expansion drove an increase in $SCA_{NBP}$ and is discussed further on. In temperate Eurasia, inversions disagree on the sign of $SCA_{NBP}$ trends and LSMs indicate weak positive trends dominated by the $CO_2$ fertilisation effect. In boreal North America, LSMs estimate $SCA_{NBP}$ trends very close to CarboScope estimates, mainly attributed to climate, whereas CAMS points to a trend close to zero because of cancelling regional trends with opposing sign (Figure 1a). CAMS and CarboScope point to increasing $SCA_{NBP}$ in temperate North

America (1.4-1.6 TgC.yr$^{-2}$), but the LSMs do not indicate any significant change (simulation S3). CAMS (which uses prior information with smaller a-priori uncertainties than CarboScope, together with a denser network) shows sharper regional differences than CarboScope, which illustrates that there are still substantial differences in the inversion at the scale of continental regions regarding $SCA_{NBP}$ trends.

Aggregated over the two latitudinal bands (Figure 2c), the MMEM indicates a dominant positive effect (increasing $SCA_{NBP}$)

of $CO_2$ fertilization both in $L_{25-40N}$ and $L_{>40N}$. In $L_{25-40N}$, the $CO_2$ effect is offset by other factors: S1 differs significantly from S2 and S3, which have lower trends of $SCA_{NBP}$. In $L_{>40N}$, the MMEM points to a positive effect of climate change in $SCA_{NBP}$ trends, thus additive to the $CO_2$ effect. The MMEM suggest a negligible contribution of LULCC to the $SCA_{NBP}$ trend in both latitudinal bands. The relative contributions of LULCC, climate and $CO_2$ however, differ between LSMs (Figure S4). Most models nevertheless agree on non-significant $SCA_{NBP}$ trends in $L_{25-40N}$ as well as on the predominant role of

$CO_2$ fertilisation and a non-significant contribution of LULCC to the trends in $SCA_{NBP}$ in $L_{>40N}$. Interestingly, models including carbon-nitrogen interactions had the weakest $SCA_{NBP}$ trends (CABLE, ISAM and LPX-Bern), excepting CLM4.5 but we cannot draw conclusions from a small sent of carbon-nitrogen models.

## 4.3 Driving processes

Validating the attribution of $SCA_{NBP}$ trends to $CO_2$, climate, and LULCC by the LSMs at large scale is challenging. How-

ever, insights into the consistency of $SCA_{NBP}$ drivers can be obtained by statistical analysis. We fitted multiple general linear regression models (GLM) to the $SCA_{NBP}$ from the inversions and the S3 MMEM over each latitudinal band using multiple predictors from the TRENDYv6 forcing datasets ($CO_2$ concentration, climate variables and changes in land-cover and management practices). For each dataset, we identified the statistical model that could best explain $SCA_{NBP}$ trends with least number of predictors (Table S2). We then tested the results from this statistical attribution for the MMEM against the corresponding





factorial estimates (see Methods). Figure 3 shows the relative contributions of the predictors (weighted by their trends) found to $SCA_{NBP}$ trends in both latitudinal bands. The coefficients of the GLM fit are shown in Figure S5.

The GLMs provide a better fit the trend of $SCA_{NBP}$ in $L_{>40N}$ (57-74% of the variance, Table S2) than for $L_{25-40N}$ (8-49% only). The GLM fit to inversions and to the MMEM identified $CO_2$ fertilisation as the most important factor explaining

(statistically) the $SCA_{NBP}$ trends in both latitudinal bands, consistent with S1 (Figures 1, S4, S5 and S6), even though the $CO_2$ fertilization effect was weaker for the GLM fit to LSMs than for inversions in region $L_{>40N}$. The statistical models for inversions and LSMs agreed on a significant negative contribution of warming in both latitudinal bands, but stronger in $L_{25-40N}$. In $L_{>40N}$, GLM models fitted to LSMs and CarboScope also point to changes in forest area contributing to increase $SCA_{NBP}$, and changes in crop area have a negative effect in $SCA_{NBP}$ from LSMs. In $L_{25-40N}$, the GLM fit to LSMs further

points to a negative contribution of wood-harvest to $SCA_{NBP}$ trends, and the fit to CAMS a weak negative effects of irrigation and fertilization. The statistical attribution of $SCA_{NBP}$ trends in LSMs is consistent with the factorial simulations, although the negative effect of temperature is only significant in $L_{25-40N}$. The key role of $CO_2$ fertilisation in the observed changes is in line with Piao et al. (2017), but our results challenge some previously proposed hypotheses to account for the increase in seasonal $CO_2$ exchange, as addressed below.

## 4.4  Confronting Hypoteses

### 4.4.1  Contribution of LULCC

Agricultural intensification and expansion occurred mainly in latitudes below 45°N (Gray et al., 2014), and inversions and LSMs reported instead a peak in the amplitude of land surface $CO_2$ exchange for latitudes above 45°N (Figure 1 and S1). Furthermore, our regional attribution identifies Eurasia as the region contributing most to increasing SCA; this region is dominated

by natural ecosystems (Figure S3) and has experienced very little land use change (Verburg et al., 2015) over the past decades. Additionally, factorial LSM simulations indicate a negligible contribution of LULCC and management to $SCA_{NBP}$ trends at latitudinal-band scale but also regionally (Figures 1c and 2).

This, though, could not in itself falsify the hypothesis that agricultural intensification is a key driver of $SCA_{NBP}$ trends, because most LSMs still do not include processes that could intensify cropland net primary productivity (NPP) over time

such as better cultivars, fertilization, irrigation. Still, management practices are not a significant predictor for GLM fitted to LSMs, but also not for inversions, excepting CAMS. CarboScope further identifies a negative effect of cropland expansion to $SCA_{NBP}$ in $L_{>40N}$ rather than a positive one, which partly challenges the contribution of cropland expansion (Gray et al., 2014) to $SCA_{NBP}$. Our results are consistent with those by Smith et al. (2014) that show that net primary productivity (NPP) generally decreased following conversion from natural ecosystems to cropland, except in areas of highly intensive agriculture

such as midwestern USA. Increasing crop productivity (intensification) could partly explain increasing $SCA_{NBP}$. However, satellite-based data for LAI (Zhu et al., 2016), NPP (Smith et al., 2016) and aboveground biomass (AGB) carbon stocks (Liu et al., 2015) (Figure S7) indicate that the increase in crop productivity accounted for only a small fraction of the hemispheric



trends in ecosystem productivity, consistent with crop productivity stagnation in Europe and Asia identified by Grassini et al. (2013).

Previous studies suggesting a large role of the green revolution in $SCA_{NBP}$ trends have focused on a longer period, starting in the 1960s. The acceleration of $SCA_{NBP}$ reported by inversions and LSMs (Table 1) concurrent with crop productivity

stagnation indicates that since the 1980s agriculture intensification is not likely to be the main driver of the increase in SCA. Even in the intensive agricultural areas in the US Midwest, CAMS estimates contrasting negative/positive trends (Figure 1a, S8). Eddy-covariance flux measurements (only for 7-13 years) in the areas of intensive agriculture in the USA show a weak relationship between trends in NBP and trends in $SCA_{NBP}$, showing mostly non-significant trends in $SCA_{NBP}$ (Figure S8).

### 4.4.2 Contribution of warming

We found that warming during the growing season had a negative effect on $SCA_{NBP}$ trends in both latitudinal bands, although this effect is uncertain for LSMs in $L_{>40N}$. Annual temperature used in the statistical models was also negatively correlated with $SCA_{NBP}$, but the correlation was only significant for CAMS. The negative relationship with growing-season temperature (T) at the mid-latitudes may be explained by warmer temperature increasing atmospheric demand for water (Novick et al., 2016) and inducing soil-moisture deficits in water-limited regions in summer (Seneviratne et al., 2010), or increased fire risk

(Peñuelas et al., 2017) that reduce the summer minimum of $SCA_{NBP}$. The negative statistical relationship found between the trend of $SCA_{NBP}$ and T in $L_{>40N}$ challenges the assumption that warming-related increase in plant productivity in high-latitudes necessarily increases the seasonal $CO_2$ exchange (Keeling et al., 1996; Graven et al., 2013; Forkel et al., 2016). Such a negative relationship, however, has also been reported by (Schneising et al., 2014) for interannual changes in the $SCA_{NBP}$ of total column $CO_2$ for 2004-2010. Yin et al. (2018) have further shown that, at latitudes between 60°N and 80°N, the relationship

between SCA NBP and T has transitioned from positive in the early 1980s, to negative in recent decades, reconciling the results by Keeling et al. (1996) and Schneising et al. (2014).

The empirical negative relationship between trends in $SCA_{NBP}$ and warming at the higher latitudes may be due to either (i) indirect negative effects of T on decomposition during the "release period"; (ii) a negative response of ecosystem productivity to warming during the "uptake period"; (iii) a stronger effect of T on total ecosystem respiration (TER) than on GPP during

the growing season. Some of these effects are counter-intuitive, because warming in high-latitudes is usually associated with longer growing-season and increased GPP (Piao et al., 2008), although a weakening of this relationship has been reported (Piao et al., 2014; Peñuelas et al., 2017).

Evidence nevertheless supports negative effects of warming on SCA trends. Temperature increase in recent decades has been associated with widespread reduction in extent and depth of snow cover (Kunkel et al., 2016) and in the number of days with

snow cover (Callaghan et al., 2011). Snow has an insulating effect, so snow-covered soil during winter can be kept at relatively constant temperatures, several degrees above the air temperature (>10°C) which promotes respiration of soil C (Nobrega and Grogan, 2007). Soils become subject to more fluctuations in temperature, and become colder, as the snow cover recedes or becomes thinner. Yu et al. (2016) reported that respiration suppression due to a reduction in snow cover in winter may account for as much as 25% of the increase in the annual $CO_2$ sink of northern forests. A decrease in respiration in response to warming





during the release period could thus decrease $SCA_{NBP}$, but the effect of growing-season temperature was stronger in our study. The expansion of vegetation in Arctic tundra, particularly shrubland, has been linked to warming trends, but also depends on soil-moisture and permafrost conditions (Elmendorf et al., 2012). Many regions of dry tundra and low arctic shrubland (Walker et al., 2005) experience summer drought or soil-moisture limitations, even though northern regions are usually considered to be

energy-limited (Greve et al., 2014). Indeed, Myers-Smith et al. (2015) found a strong soil-moisture limitation of the (positive) sensitivity of shrub growth to temperature in summer, possibly associated with the limitation of growth due to drought and/or with reduced growth and dieback due to standing water during thawing. CAMS indicates a decrease in $SCA_{NBP}$ in eastern regions in boreal North America (Figure 1a), where Myers-Smith et al. (2015) reported negative sensitivity of shrub-growth to temperature. The coarse network and large correlation lengths used by CarboScope do not allow such regional contrasts to

be resolved. Most process-based models lack a detailed representation of processes described above – e.g. a realistic effect of snow insulation on soil temperatures, soil freezing and thawing (Koven et al., 2009; Peng et al., 2016; Guimberteau et al., 2017) – potentially overestimating the net sink response to temperature changes (Myers-Smith et al., 2015). Moreover, soil-moisture limitation due to temperature increase could also contribute to decrease TER by limiting microbial activity, which is currently not simulated in most LSMs. This may in turn explain why LSMs underestimate the negative effect of temperature

in $SCA_{NBP}$ in the high-latitudes compared to CAMS (Figure 3 and S4).

### 4.4.3 Evaluating model biases

Wenzel et al. (2016) proposed that the observed sensitivity of $SCA_{NBP}$ to $CO_2$ was an emergent constraint on future terrestrial photosynthesis, but their study focused on simulations by an earth-system model that excluded the effects of climate change (i.e. the radiative feedback of $CO_2$ to climate was not considered). Our results are consistent with a strong increase in the peak

uptake due to the effect of $CO_2$ fertilisation driven by gross primary production (GPP) as proposed by Wenzel et al. (2016). The negative effect of temperature in our study (Figure 3), although weaker than the positive effect of absolute $CO_2$ concentration, suggested that warming partly cancelled out the increase in $SCA_{NBP}$ expected from the effect of fertilisation alone. We propose that other processes partly control $SCA_{NBP}$ trends linked to reduced decomposition under lower snow-cover (Yu et al., 2016) or to emerging limitations to growth in response to water-limitation (Elmendorf et al., 2012; Myers-Smith et al.,

2015). Additionally, while the sensitivity of productivity to the $CO_2$ fertilisation effect is expected to decrease, whereas the control of respiration by temperature should increase nonlinearly (Piao et al., 2014, 2017; Peñuelas et al., 2017), suggesting a progressively dominant (negative) influence of warming on $SCA_{NBP}$. The degree of such an offset would likely depend on the thresholds of soil temperature and water limitation that are complex and thus difficult to assess and require process-based modelling. Our results imply that future constraints of productivity based only on the $CO_2$ effect (as in Wenzel et al. (2016))

may overestimate future GPP.

We evaluated whether the differences between the observed $SCA_{NBP}$ trends (significant only in $L_{>40N}$) and those simulated by the LSMs could be associated with the modelled sensitivities to atmospheric $CO_2$ concentration ($CO_2$) and growing-season temperature (T) in $L_{>40N}$ (Figure 4). In Figure 4, only models with a too small sensitivity of $SCA_{NBP}$ to T produce



a realistic trend of $SCA_{NBP}$. In contrast, the models indicating sensitivities to T and $CO_2$ more similar to those estimated by the inversions tend to underestimate the trend in SCA.

Why are the LSM sensitivities of SCA to T positively correlated with their long-term $SCA_{NBP}$ trend (Figure 4), even though $CO_2$ is a stronger driver of the simulated $SCA_{NBP}$ trend (Figure 3)? We found a clear relationship between the model bias in the trend of $SCA_{NBP}$ and the sensitivity to $CO_2$ fertilisation in S3 (in line with Wenzel et al. (2016)), but we also found a compensatory effect, where models that overestimate the sensitivity of SCA to T tend to underestimate the sensitivity to $CO_2$ and vice-versa. LSMs tend to overestimate (underestimate) sensitivity of $SCA_{NBP}$ to T ($CO_2$), compared to the observation-based constraints from inversions. LSMs often compensate too strong (or too weak) simulated water-stress or temperature sensitivity by adjusting photosynthesis parameters (that control $CO_2$ fertilization) during model optimization to match the observed net terrestrial sink. This compensatory effect has previously been reported by Huntzinger et al. (2012) for the mean terrestrial sink; we find that it could also affect the trends in seasonal $CO_2$ exchange.

We argue that the trend of $SCA_{NBP}$ can differ between models due to: a) differences in their NPP response to T and $CO_2$; b) differences in turnover times of short-lived C pools by which increased NPP is coupled to increased winter decomposition; (c) phase shifts between GPP and ecosystem respiration. The latter may be associated with errors in the phase and amplitude of simulated ecosystem respiration, arising from factors such as: (i) representing soil carbon stocks as pools with discrete turnover times and associated effective soil depths Koven et al. (2009) (ii) neglect of seasonal acclimation effects on autotrophic and heterotrophic respiration. The sensitivities of NPP to $CO_2$ and T between models are strongly and consistently correlated with the compensatory effect of the model parameterisations (Figure S9), but we find no clear relationship between the biases of the modelled $SCA_{NBP}$ trend and the sensitivity of NPP to T (Figure S10), suggesting a key role of respiration. Indeed, the models with $SCA_{NBP}$ trends closer to observations tend to be associated with a lower sensitivity of ecosystem respiration to growing-season temperature (Figure S4c). Too large turnover of short-lived pools in a model should produce too small increase of the $SCA_{NBP}$ amplitude (i.e. increased respiration during the "uptake period" followed by too little during the "release period") for a given sensitivity of NPP to $CO_2$ or climate. A recent study by Jeong et al. (2018) has reported that ecosystem carbon-cycle models (not used in this study) underestimated changes in carbon residence times in northern Alaska. The evaluation of the effect of model turnover times in $SCA_{NBP}$ requires a deeper analysis of transfers between litter and soil organic carbon pools and can be verifiable in future simulations.

## 5 Conclusions

Based on our assessment of atmospheric observations and most advanced land-surface model simulations the most likely explanation of the seasonal cycle of atmospheric $CO_2$ at high latitudes is the $CO_2$ fertilization of photosynthesis in unmanaged high latitude ecosystems, especially in the Eurasian Boreal forests. Our study further points to key processes that need to be developed to better simulate NBP responses to changing climate, especially to Arctic warming, in particular productivity limitations and the decomposition terms. Our results indicated that the signal of the $SCA_{NBP}$ trend contains valuable information for the turnover times of short-term pools, which await further investigation.



*Author contributions.* A.B. and P.C. designed the study, conducted the analysis and wrote the manuscript. A.P.B., F.C., C.R., F.M., M.F-M., J.P., S.L.P. W.K.S., X.W., Y.Y. and Z.Z. contributed with expert knowledge during the development of the study. S.S and P.F coordinated the TRENDY simulations and maintained the TRENDYv6 data. F.C. and C.R. developed the atmospheric inversion datasets and contributed to the analysis of inversions. V.H., E.K.,A.K.J.,S.L.,D.L.,J.E.M.S.N., P.P., B.P. and D.Z. performed the TRENDYv6 simulations. All authors
5   contributed to the writing of the manuscript.

*Competing interests.* We have no competing interests.

*Acknowledgements.* This work was partly supported by the European Space Agency Climate Change Initiative ESA-RECCAP2 project (ESRIN/ 4000123002/18/I-NB).



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





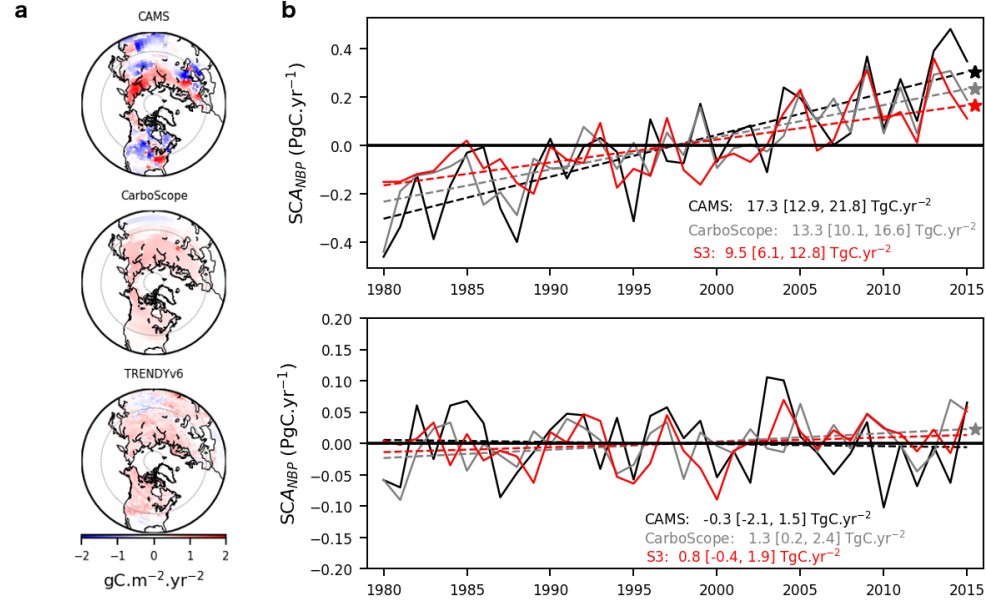

**Figure 1.** Variability of seasonal-cycle amplitude and trends from the inversions and LSMs. (a) Geographical distribution of $SCA_{NBP}$ trends from the inversions (CAMS and CarboScope) and the multi-model ensemble (MME) mean from TRENDYv6 simulation S3 (all forcings). Both inversions estimated predominantly positive trends in $SCA_{NBP}$ >40°N (Figure S1), so we defined two latitudinal bands, $L_{>40N}$ and $L_{25-40N}$, for flux aggregation. (b) Aggregated $SCA_{NBP}$ time-series estimated by the inversions (CAMS in black and CarboScope s76 in grey) and S3 MME mean (red). The dashed lines indicate the linear fits used to calculate the slopes of the trends (corresponding colours), and the slopes and confidence intervals (95%) are provided.





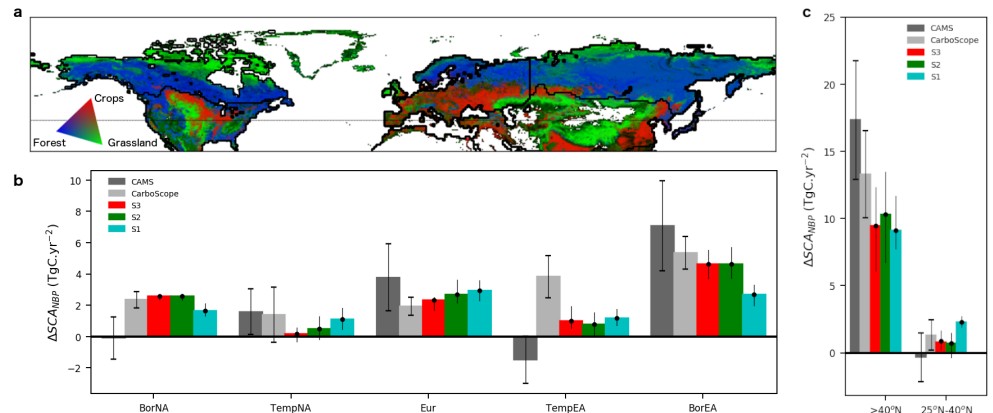

**Figure 2.** Regional distribution of the dominant land-cover types and $SCA_{NBP}$ trends. (a) Land-cover map averaged over the study period for the three main land-cover classes (forest/shrubland, grassland, and cropland) based on ESA-CCI annual land-cover data (1992-2015 average); (b) The continental regions correspond to the regions defined by Baker et al. [2006] and are delimited by bold lines: boreal and temperate North America (BorNA and TempNA), Europe (Eur), and boreal and temperate Eurasia (BorEA and TempEA); (c) Comparison of the $SCA_{NBP}$ trends from the inversions to the trends estimated by the LSM experiments: S3, S2 (no LULCC), and S1 (no LULCC and no climate change). The bars for the inversions and LSMs indicate the average trend over each latitudinal band. The error bars for the inversions indicate the 95% confidence levels for the trend values, and the vertical lines for the LSMs indicate inter-quartile ranges of the MME. The 95% confidence interval for the MME mean was also calculated (see Methods).



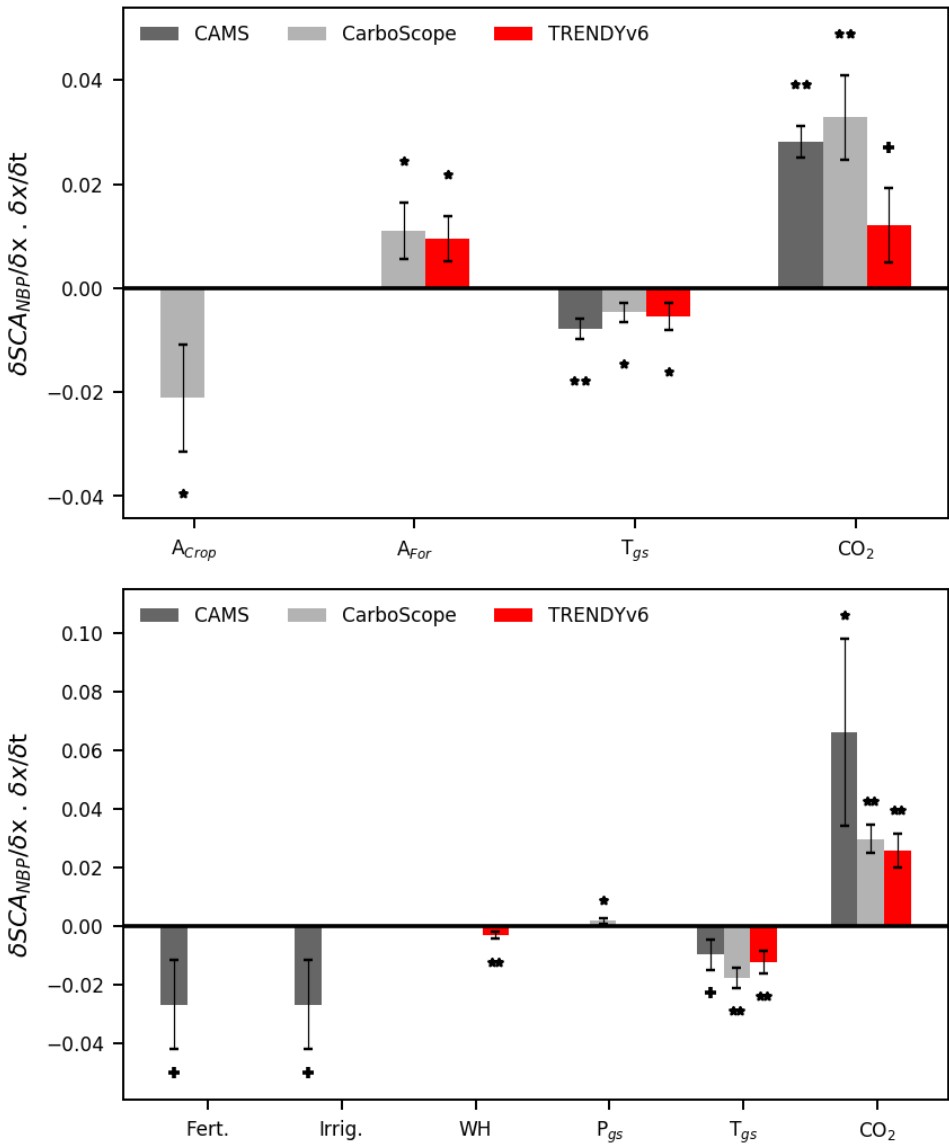

**Figure 3.** Statistical attribution of drivers of $SCA_{NBP}$ estimated by the inversions and LSMs. The main drivers of $SCA_{NBP}$ are presented for (a) $L_{>40N}$ and (b) $L_{25-40N}$ and are calculated as the product of the coefficients of a general linear model fit on $SCA_{NBP}$ using a number of predictors (normalised) and their corresponding trends. Fertilization, irrigation, wood harvest, growing-season precipitation, growing-season temperature, atmospheric $CO_2$ concentration were tested as predictors, and the best fit was chosen for each dataset: CAMS (dark grey), CarboScope s76 (light grey), and the MMEM (red). The bars indicate the contribution of each predictor to the trend in $SCA_{NBP}$, error bars indicate the corresponding 95% confidence intervals, and the symbols indicate significant MRLM fits (two, one asterisks and crosses, $p<0.01$, $p<0.05$ and $p<0.1$ respectively).




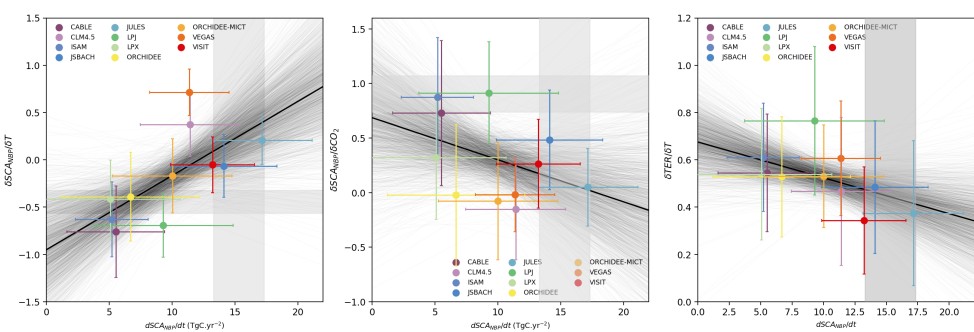

**Figure 4.** Emerging relationships between LSM sensitivities to climate and $CO_2$ and their SCA trends. The $SCA_{NBP}$ trend for $L_{>40N}$ estimated by each inversion (grey intervals) and corresponding responses of $SCA_{NBP}$ to (a) T and (b) $CO_2$ (as calculated in Figure 3 but considering the scores of the regression only, shown in Figure S4) are compared to the results from individual models (simulation S3, coloured markers). The shaded areas indicate the inversion ranges, and the distribution of the grey lines shows uncertainty in the relationship between each pair of variables.