# Peer review of "Contrasting effects of CO2 fertilisation, land-use change and warming on seasonal amplitude of northern hemisphere CO2 exchange"

_Atmospheric Chemistry and Physics, 2019_

## Referee Comment (RC1) · Anonymous Referee #1 · 30 Apr 2019

This paper investigated the relative importance of individual contributors to trends and drivers of the seasonal-cycle amplitude (SCA) in northern high latitudes using two atmospheric inversions and land-surface models. They found the most likely explanation of the trend of SCA at high latitudes is the CO2 fertilization of photosynthesis, rather than LULCC. Although I see the value of publishing, I am concerned about the definition of SCA and reliability of results.

The SCA of atmospheric CO2 should be the difference between the peak and trough values of the cumulative CO2 in a year. But the definition of SCA in this manuscript is the difference between peak uptake and trough of NBP. The sum of NBP during the

growing season is related to the SCA of atmospheric CO2 while the difference between peak uptake and trough of NBP may be not.

It will be clearer if the Result and discussion can be separated into two part alone. The key finding is CO2 fertilization drive the SCA trend, but more discussion and speculation focused on warming.

Page 2 Line 8, how many are the relative effects of CO2 fertilization and warming on SCA, respectively? Page 5 line 8 and line 28 typos Page 5, why did you use ESA-CCI Land-Cover data set for the analysis of satellite-based vegetation data sets? what are the problems if LUH2 was used for the analysis of satellite-based vegetation data sets? Page7 line4, figure S was missed The size of Fig1.a is too small to see them clearly. Also for figure 4. Page 7 line 15, how did you know the breakpoint in the north of 40°N? Page 8, The patterns of SCANBP trends from the LSM were not consistent with that of CAMS at the pixel scale. The attribution analysis based on LSMs is not very convincing. Page 9 line 29-34, these sentences should be moved into Method.
* * *

---

## Author Comment (AC1) · 28 May 2019

RC1: This paper investigated the relative importance of individual contributors to trends and drivers of the seasonal-cycle amplitude (SCA) in northern high latitudes using two atmospheric inversions and land-surface models. They found the most likely explanation of the trend of SCA at high latitudes is the CO2 fertilization of photosynthesis, rather than LULCC. Although I see the value of publishing, I am concerned about the definition of SCA and reliability of results. The SCA of atmospheric CO2 should be the difference between the peak and trough values of the cumulative CO2 in a year. But the definition of SCA in this manuscript is the difference between peak uptake and

trough of NBP. The sum of NBP during the growing season is related to the SCA of atmospheric $CO_2$ while the difference between peak uptake and trough of NBP may be not.

AR: Most previous studies indeed have analyzed trends in SCA of atmospheric $CO_2$ concentrations (Graven et al., 2013; Forkel et al., 2016; Thomas et al., 2016; Zhu et al., 2016; Piao et al., 2017; Yin et al., 2018). However, to attribute changes in the seasonal amplitude of atmospheric $CO_2$ to specific processes it is necessary to look at net surface fluxes as a function of changes in primary productivity and respiration. Moreover, quantifying bias in $CO_2$ concentration at a given site from a bias in land-surface model (LSM) simulated fluxes is difficult, since the biases can be affected by many other factors such as transport model characteristics, forcing data used, etc. As discussed in the Introduction (P2 L 29 to P3 L2), atmospheric inversions might partly tackle this issue by limiting the space of surface fluxes that are consistent with the atmospheric $CO_2$ concentration measured at several sites. Moreover, when aggregated at large spatial scales, the annual amplitude of NBP is related with the amplitude of the concentration (although this relationship is complicated by atmospheric transport, to the first order, the SCA of concentration should roughly be the integral of the flux). Such an approach has for example been used by Welp et al. (2016) for boreal ecosystems. Finally, here we compare results from two inversion systems and results from sensitivity runs from CarboScope forced with different inputs and using different parameters. This allows further insight about the range of SCANBP values that can still be compatible with in-situ atmospheric $CO_2$ measurements. By doing this, we believe we can provide a fair evaluation of the ability of LSMs to capture changes in the seasonal amplitude of NBP (and $CO_2$) in the Northern Hemisphere.

Welp, L. R., Patra, P. K., Rödenbeck, C., Nemani, R., Bi, J., Piper, S. C., and Keeling, R. F.: Increasing summer net $CO_2$ uptake in high northern ecosystems inferred from atmospheric inversions and comparisons to remote-sensing NDVI, Atmos. Chem. Phys., 16, 9047-9066, https://doi.org/10.5194/acp-16-9047-2016, 2016.

RC2: It will be clearer if the Result and discussion can be separated into two part alone. The key finding is CO2 fertilization drive the SCA trend, but more discussion and speculation focused on warming.

AR: The results and discussion sections will be separated in the revised version. We believe that our finding that warming has a negative effect on SCANBP is also a key finding of this study, and the one deserving more explanation. The effect of CO2 fertilization in increasing CO2 uptake is well understood from a physiological point of view, while the effects of temperature on SCANBP are complex and, in this case, counter-intuitive. In fact, earlier studies pointed for a positive effect of warming on SCA because of earlier onset of the growing-season or increase growth at higher latitudes (Keeling et al., 1996; Forkel et al. 2016). The negative effect of warming we find seems though to be supported by studies covering a more recent period (Schneising et al., 2014; Peñuelas et al., 2017; Yin et al., 2018), although the mechanisms behind were not discussed. Here we try to understand this by analyzing the link between T and GPP and TER simulated by models. The fact that models show biases in their simulated sensitivity of SCA and TER to T indicates that certain processes might be missing. We point to some processes that might explain these biases, based on published research, rather than speculation. To evaluate whether these processes can or cannot explain the biases, these would need to be included in model simulations, which is beyond the goals of our study.

RC3: Page 2 Line 8, how many are the relative effects of CO2 fertilization and warming in SCA, respectively?

AR: This is discussed in the following paragraphs of the introduction.

RC4: Page 5 line 8 and line 28 typos

AR: Corrected.

RC5: Page 5, why did you use ESA-CCI Land-Cover data set for the analysis of

satellite-based vegetation data sets? What are the problems if LUH2 was used for the analysis of satellite-based vegetation data sets?

AR: We used ESA-CCI Land-Cover because it is a purely remote-sensing based land-cover dataset, while LUH2v2h is partly based on HYDE3.1, which in turn uses FAO data for cropland extent. However, since LUH2v2h is used to force the LSMs, it is true that a comparison with this dataset should also be made. We have now compared the results in Fig. S6 using LUH2v2h. We compare trends in LAI, NPP and AGB for the LUH2v2h classes cropland, forest and non-forest natural vegetation (which should include shrublands and natural grasslands), for the period 1982-2015, for latitudes north of 40oN. As in our results with ESA-CCI Land-Cover, forests contribute the most to LAI, NPP and AGB increase.

In the revised version of the manuscript we can add these results as a second panel in Figure S6 (Fig. 1 below).

RC6: Page7 line4, figure S was missed

AR: It should read S5, it has been accordingly corrected.

RC7: The size of Fig1.a is too small to see them clearly. Also for figure 4.

AR: The figures will be improved.

RC8: Page 7 line 15, how did you know the breakpoint in the north of40âŮęN?

AR: It is the point north of which the two inversions agree on a significant sign of SCANBP trends. This will be clarified in the text in a revision of the manuscript.

RC9: Page 8, The patterns of SCA NBP trends from the LSM were not consistent with that of CAMS at the pixel scale.

AR: Inversion fluxes are highly uncertain at pixel-scale (discussed in P6 L12, P8 L30-31) and should not be directly compared with pixel-scale LSM fluxes, especially in regions where there are sparse atmospheric $CO_2$ measurements. The large-scale

spatial distribution of SCANBP is shown in Fig. 1 to illustrate the distinct results from the two observation-based datasets (which underlines the problems of relying on single atmospheric-transport models to forward-transport fluxes). The sentence will be reformulated in a future revision of the MS.

RC10: The attribution analysis based on LSMs is not very convincing.

AR: We do not necessarily agree with the reviewer, especially because the reviewer has not identified specific weaknesses in our analysis or conclusions. Attribution of changes in SCA (or NBP) to CO2, climate and LUC can be made using statistical methods or performing modelling experiments. For observation-based data, statistical attribution is the only option, and we try to do disentangle the effect of each term from the others by fitting statistical models with different numbers and combinations of predictors. Process-based models, on the other hand, allow us to evaluate individual processes that may be contributing to the observed patters by running simulations in which the LSMs are forced with only one, two or more factors. The effect of CO2, climate and LUC can then be diagnosed by the differences in resulting SCA between experiments. We would like to note that model-based attribution is actually the approach followed by most studies analyzing trends in NBP or in SCA (Graven et al., 2013; Forkel et al., 2016; Thomas et al., 2016; Zhu et al., 2016; Piao et al., 2017). The difficulty with the attribution by models is that it cannot easily be validated, as discussed in the manuscript. Therefore, we compare: (i) the process attribution from factorial simulations, (ii) the regional statistical attribution based on inversion fluxes, (iii) the statistical attribution based on LSMs fluxes from S3 and (iv) the statistical attribution based on the differences between factorial simulations. This allows testing the statistical attribution, and allows comparing the results from observation-based data with simulated data. To the best of our knowledge this is the most robust way to perform such attribution, and it has not been done in other studies (which have relied mainly on factorial simulations and did not compare with observation-based data). While each attribution approach may have their respective limitations, the fact that our regional attribution identifies the

Eurasian Boreal forest as a major contributor to SCA, and our process attribution identifies $CO_2$ fertilization of Eurasian Forests as the mechanism, provides more support for the natural vegetation hypothesis than the agricultural intensification hypothesis.

RC11: Page 9 line 29-34, these sentences should be moved into Method

AR: The sentences were redundant as this was already discussed in the Methods, so they were removed.

Please find a PDF version of this reply attached.

Please also note the supplement to this comment:
https://www.atmos-chem-phys-discuss.net/acp-2019-252/acp-2019-252-AC1-supplement.pdf

―――――――――――――――――

[Figure]

[Figure]

**Fig. 1.**

---

## Referee Comment (RC2) · Anonymous Referee #2 · 3 Jul 2019

The increase of the seasonal-cycle amplitude (SCA) of CO2 has been long researched. This study utilized the inversions and LSM simulations to research the main drivers of the enhanced SCA, and pointed out that the effects of CO2 fertilization and warming on SCA have the contrasting effects. However, I have a big concern that whether the GLM can give us the reliable result. It can be a good prediction model but not for causal analysis, especially the predictors here you used (eg. Temperature and CO2) have the high correlations. So (a) I think you should show a figure that makes a direct comparison between the statistical decomposition (CO2, Tgs, ..) and factorial simulations (S1,S2-S2, S3-S2) upon TRENDY S3 NBP, not in the form of your Figure S6 (slope). (b) In your Figure S6b, we can focus on the green bar which represents the

climate effect only. But after your MLRM fit, we can find that the WH and CO2 also have the significant effect. (c) We can see the climate effect is positive in model experiments in Figure 2c, but temperature effect is negative in statistical analysis in Figure 3a. So what's the matter? These phenomena show that the explanations should be cautious.

Details: (1) Abstract Line 4: 'from and 11 state-of-the-art' remove the and (2) In introduction, the last two paragraphs can be place into Section Data (3) Page 7 line 4 'The coefficients from the GLM fit for each datasets are shown in Figure S' maybe Figure S5; The last sentence in next paragraph should be 'Figure S6' (4) Page 7, line 14. 'strong decreasing trends for mid-latitudes'-only CAMS shows (5) Page 9, line 12-13 'In boreal North America, LSMs estimate SCAnbp trends very close to CarboScope estimates, mainly attributed to climate' not only to climate, CO2 effect even stronger. (6) Page 11, line 23 '(i) indirect negative effects of T on decomposition during the "release period"' why negative effects? It seems a positive effect, because warmer temperature can result in more release of C by respiration, which can enlarge the SCA.

---

## Author Comment (AC2) · 31 Jul 2019

RC1: The increase of the seasonal-cycle amplitude (SCA) of CO2 has been long researched. This study utilized the inversions and LSM simulations to research the main drivers of the enhanced SCA, and pointed out that the effects of CO2 fertilization and warming on SCA have the contrasting effects. However, I have a big concern that whether the GLM can give us the reliable result. It can be a good prediction model but not for causal analysis, especially the predictors here you used (eg. Temperature and CO2) have the high correlations. So (a) I think you should show a figure that makes a direct comparison between the statistical decomposition (CO2, Tgs, ..) and factorial simulations (S1,S2-S2, S3-S2) upon TRENDY S3 NBP, not in the form of your Figure S6 (slope).

AR1: We agree with the reviewer that attribution simply based on the statistical GLM fit would be insufficient for causal analysis. The rationale for combining the attribution by LSMs with that of the GLM is the following:

Trends in SCA can be quantified from observation-based datasets (here the inversions) or from simulations of net land-atmosphere fluxes by LSMs. As they are based on multiple in-situ measurements, the former should provide a reference (and respective uncertainty) for evaluating the LSMs results. The only option to identify drivers of SCA from inversions is statistical attribution. LSMs, on the other hand, allow separating the contribution of each term through the different factorial simulations. However, the attribution by LSMs cannot be easily validated, which is especially problematic given that LSMs underestimate the trends in SCA at latitudinal scale, and show regional mis-matches with inversions. Thus, we compare: (i) the process attribution by LSMs (S1, S2 and S3), (ii) the statistical attribution based on inversion fluxes, (iii) the statistical attribution based on LSMs fluxes from S3 (directly comparable to the inversion results) and (iv) the statistical attribution based on the differences between factorial simulations (cross-evaluation of (i) and (iii)). We believe this is a robust way to evaluate attribution from both inversions and LSMs, and it has not been done in other studies (which have relied mainly on factorial simulations and did not compare with observation-based data). We agree with the referee that it is more meaningful to show the direct decomposition, rather than the sensitivities. We have therefore updated Figure S6, as shown in Figure R1. The discussion around Fig. S6 has accordingly been changed.

RC2: (b) In your Figure S6b, we can focus on the green bar which represents the climate effect only. But after your MLRM fit, we can find that the WH and CO2 also have the significant effect.

AR2: In the revised version of Fig. S6, the climate effect is only significant for Tgs and HW, but very small for the latter. On the other hand, the positive $CO_2$ fertilization effect is clearly dominant and found in S1. We acknowledge that the discussion of Fig. S6 should be improved, and have addressed this issue in the answer to the previous comment.

RC3: (c) We can see the climate effect is positive in model experiments in Figure 2c, but temperature effect is negative in statistical analysis in Figure 3a. So what's the matter? These phenomena show that the explanations should be cautious.

AR3: The climate effect in Figure 2c encompasses changes in all the relevant variables for ecosystem productivity (temperature, radiation, precipitation, wind) throughout the year, while Figure 3a shows only the effect of growing-season temperature (i.e. Tgs). Therefore, it is possible for the effect of the combined climate changes to be positive, but for the effect of

Tgs to be negative. This is the reason why we compare the statistical attribution from inversions to that of LSMs.

RC4: Details: (1) Abstract Line 4: 'from and 11 state-of-the-art' remove the and
AR4: Corrected.

RC5: (2) In introduction, the last two paragraphs can be place into Section Data
AR5: The information in these two paragraphs was redundant. We merged the key points with the Methods sections and removed the parts that were repeated elsewhere.

RC6: (3) Page 7 line 4 'The coefficients from the GLM fit for each datasets are shown in Figure S' maybe Figure S5;The last sentence in next paragraph should be 'Figure S6'
AR6: Both references to the figures have been corrected.

RC7: (4) Page 7, line 14. 'strong decreasing trends for mid-latitudes'-only CAMS shows
AR7: We have reformulated the sentence:
"[…] even though CAMS shows heterogeneous patterns in North America with strong decreasing trends for mid-latitudes (Figure 1a, S1)."

RC8: (5) Page 9, line 12-13 'In boreal North America, LSMs estimate SCAnbp trends very close to CarboScope estimates, mainly attributed to climate' not only to climate, CO2 effect even stronger.
AR8: Thank you for noting the error. We have corrected the sentence to:
"[…] mainly attributed to $CO_2$ followed by climate […]"

RC9: (6) Page 11, line 23 '(i) indirect negative effects of T on decomposition during the "release period"' why negative effects?   It seems a positive effect, because warmer temperature can result in more release of C by respiration, which can enlarge the SCA.
AR9: Indeed, higher temperatures are expected to increase respiration both during the "net uptake period" and the "net release period". However, the effects of temperature are seasonally-dependent as exemplified below. In the left panel of figure R2, increasing T might increase GPP during the uptake period (having a positive effect on SCA), but at the same time increase maintenance respiration and decomposition in the uptake period, and leading to an increase in C available for decomposition during the release period. The latter effect would decrease SCA during the growing season, and increase SCA in the release period. On the other hand (middle panel), T might increase water-stress and contribute to decrease growing-season GPP (decreasing SCA). Consequently, maintenance respiration could be decreased in the growing-season (offsetting part of the GPP decrease effect), but also in the release-period (contributing to decrease SCA). Finally, in snow-covered regions, warming might contribute to reduce the snow-cover, which in turn might imply more TER during the growing-season but also increased GPP (left panel), but also have delayed effects during the release period, by reducing the insulation cover and inhibiting TER during the release period, which would contribute to a decrease in SCA (right panel). This figure and corresponding discussion have been added to the Supplementary Material.

[Figure]

Figure R1 (new Figure S6): Factorial verification of the drivers in TRENDY S3 for (a) >40°N and (b) 25-40°N. The MLRM fit to the partial fluxes for the effects of LULCC (S3-S2, red), climate (S2-S1, green), and $CO_2$ fertilisation (S1, cyan). Results should be compared to those in Figure 3. The significant predictors in the GLM fit to the LSMs in S3 should be detected in the corresponding factorial simulations. It should however be noted that management and fertilization are already included in S1 and S2 for some models. The difference between S3 and S2 (LULCC effects) mainly suggest LULCC processes and does not identify the effect of $CO_2$, except if there are interactions between the $CO_2$ fertilization effect and LULCC emissions (e.g. higher emissions from deforestation because of higher C-stocks). The effect of $CO_2$ is identified mainly by the difference in S1-S0 and S2-S1, possibly due to synergies between $CO_2$ fertilisation and climate change. The effect of temperature should be evident in the difference between S2 and S1 (effects of climate), consistently found in $L_{25-40N}$.

[Figure]

Figure R2: Conceptual scheme of the impacts of warming in SCA.

---

## Author Response (AR2)

**MS acp-2019-252**
**Authors' response to Editor**

We thank the Editor for pointing out the aspects of our revision that could have been made clearer for the general readers. We have now included these clarifications in the manuscript, following the Editor's remarks. Please find below a point by point description of the changes made to the manuscript, and a new revised document with tracked changes (compared to the discussion manuscript).

EC1:
- Reviewer 1:
- RC1: This is the major concern of Rev1 but you appear to have made no changes to the manuscript, despite a lengthy response to the point. Consider that your readers will often think the same thing as the reviewer and please make sure to thoroughly address this point in the article text. A discussion of how results from this study, with a different SCA definition, can be compared to previous studies would be especially helpful.
AR: We have added one paragraph in the Introduction to clarify this point and added the reference to Welp et al. (2016). The paragraphs now read (revised sentences underlined):

> "[…] Piao et al. (2017) further reported that $CO_2$ fertilisation and climate change drove the increase in SCA for sites >50°N, but that at mid-latitude sites land use, oceanic fluxes, fossil-fuel emissions, as well as trends in atmospheric transport may have contributed to the SCA trends.
> Attributing changes in the seasonal amplitude of atmospheric $CO_2$ to specific processes requires analysing net surface fluxes as a function of changes in gross fluxes (photosynthesis, respiration, disturbance), which LSMs can provide. However, quantifying bias in $CO_2$ concentration at a given site from a bias in land-surface model (LSM) simulated fluxes is difficult, since the biases can be affected by many other factors such as transport model characteristics, forcing data used, among others.
> Atmospheric inversions provide a consistent framework for assimilating in-situ $CO_2$ concentration observations to estimate net $CO_2$ surface fluxes while accounting for errors in the prior fluxes and for some errors in the ATM (Peylin et al., 2013). At large spatial scales, the trends in SCA can be related with trends in the seasonal amplitude of $CO_2$ fluxes (i.e. SCA of NBP, SCANBP). Such an approach has been used to analyse trends in net $CO_2$ uptake in boreal regions (Welp et al., 2016).
> The spatiotemporal distribution of terrestrial and oceanic surface fluxes estimated by inversions provides thus direct insight […]"

EC2: - Similarly for RC2, 5 and 10: If you have a lengthy response to the reviewer's comment, please include the main points of it in the revised manuscript. Submit an author's response that clearly highlights the changes made and shows how you have clarified such that this point will be clear to future readers.
AR:
Regarding RC2: In the original version of the MS the header Results and Discussion was misleading, since the sections "Large Scale Patterns", "Regional Attribution" and "Process Attribution" only presented results, while the section "Confronting Hypotheses" and "Evaluating Model Biases" correspond to a discussion of the results: implications for

previously proposed hypotheses, and discussion about why LSMs may present biases compared to inversions. We have therefore corrected the section titles, but do not believe that the sections should be reorganized. Regarding the second aspect of RC2, we had included a new Figure (S9) with a conceptual scheme of the effects of warming on $SCA_{NBP}$. The discussion was added in the Supplement rather than in the main text mainly due to length considerations. Nevertheless, we agree that the discussion can be clarified in the MS, which now reads (changes underlined):

> "[…] the relationship between SCA NBP and T has transitioned from positive in the early 1980s, to negative in recent decades, reconciling the results by Keeling et al. (1996) and Schneising et al. (2014).
> In Figure S9 we present a conceptual scheme of the impacts of warming in SCANBP through its component fluxes. Generally, warming in high-latitudes has been associated with longer growing-season and increased GPP (Piao et al., 2008), which would contribute to increase SCANBP through increased productivity during the "uptake period" and increased decomposition (due to more litter) during the "release period". However, a weakening of this relationship has been reported (Piao et al., 2014; Peñuelas et al., 2017). Other processes can, though, contribute to the negative relationship between $SCA_{NBP}$ and T reported here and in other studies (Figure S9). The empirical negative relationship between trends in $SCA_{NBP}$ and warming at the higher latitudes may be due to either (i) a stronger effect of T on total ecosystem respiration (TER) than on GPP during the "uptake period"; (ii) a negative response of ecosystem productivity to warming during the "uptake period"; (iii) indirect negative effects of T on decomposition during the "release period".

Regarding RC5: We have now added a reference to the comparison between ESA-CCI LandCover and LUHv2h in the Data section:

> "LUH2v2h was used for the statistical analysis of inversion and the LSM drivers (because it was the data set used to force the models), and ESA-CCI data were used for the analysis of satellite-based vegetation data sets, and results were additionally compared with LUH2v2h."

And in the Discussion section:

> "However, satellite-based data for LAI (Zhu et al., 2016), NPP (Smith et al., 2016) and aboveground biomass (AGB) carbon stocks (Liu et al., 2015) for different land-cover classes from ESA-CCI LC (Figure S7) indicate that the increase in crop productivity accounted for only a small fraction of the hemispheric trends in ecosystem productivity, consistent with crop productivity stagnation in Europe and Asia identified by Grassini et al. (2013). We also compared the trends in remote-sensing variables for land-cover classes from LUH2v2h, with similar results."

Regarding RC10: We agree that a description of the rationale behind our approach could be better stressed in the manuscript. Therefore, we have added the following paragraphs at the end of the Introduction:

> "Trends in $SCA_{NBP}$ from the inversions are based on multiple in-situ measurements and therefore provide a reference (and respective uncertainty) for evaluating the regional attribution by LSMs. Regarding process attribution, LSMs allow separating the contribution of different drivers through factorial simulations. However, the attribution by LSMs cannot be easily validated, which is especially problematic given that LSMs underestimate the trends in SCA at latitudinal scale (Thomas et al., 2016).

Thus, we compare: (i) the process attribution by LSMs (as e.g. in Thomas et al. (2016); Piao et al. (2017))), (ii) the statistical attribution based on inversion fluxes, (iii) the statistical attribution based on LSMs fluxes, directly comparable to the inversion results and (iv) the statistical attribution based on the differences between factorial simulations (cross-evaluation of (i) and (iii)).

Our approach allows thus to constrain $SCA_{NBP}$ trends at hemispheric and regional scales from both top-down (inversions) and bottom-up (LSMs) methods and to evaluate the process attribution by LSMs using top-down estimates of $SCA_{NBP}$."

EC3: - Reviewer 2:
- RC1: It sounds like you have made some changes here though perhaps not enough based on the strength of the reviewers concern. Again, make sure all info is included in the revised article, and please present all changes made point by point in the author response.
AR: This has now been corrected, please see reply to EC2, regarding RC10 of Reviewer #1.

EC4: RC3: The reviewer was clearly confused here by the presentation and didn't notice the different scopes of these figures. Don't just explain in the response, clarify in the article.
AR: We agree that this can be clarified in the manuscript. We have now changed Section 5.1.2 as follows:

"We found that warming during the growing season had a negative effect on $SCA_{NBP}$ trends in both latitudinal bands, although this effect is uncertain for LSMs in $L_{>40N}$. Annual temperature used in the statistical models was also negatively correlated with $SCA_{NBP}$, but the correlation was only significant for CAMS.

The negative relationship with growing-season temperature (T) at the mid-latitudes may be explained by warmer temperature increasing atmospheric demand for water (Novick et al., 2016) and inducing soil-moisture deficits in water-limited regions in summer (Seneviratne et al., 2010), or increased fire risk (Peñuelas et al., 2017) that reduce the summer minimum of $SCA_{NBP}$. This negative effect of temperature can explain the negative contribution of climate to the simulated $SCA_{NBP}$ trends in $L_{25-40N}$ given by the factorial simulations (Figure 2c).

The negative statistical relationship found between the trend of $SCA_{NBP}$ and T in $L_{>40N}$ challenges the assumption that warming-related increase in plant productivity in high-latitudes necessarily increases the seasonal CO2 exchange (Keeling et al., 1996; Graven et al., 2013; Forkel et al., 2016). Although the MMEM shows a small positive contribution of climate in $L_{>40N}$, LSMs diverge on the contribution of climate in this latitudinal band (Figure S4). Moreover, the factorial simulations in Figure 2c allow evaluating the impact of changes in all climate variables (e.g. also rainfall and radiation), in addition to temperature. A negative relationship between T and SCA has, though, also been reported by (Schneising et al., 2014) for interannual changes in the $SCA_{NBP}$ of total column $CO_2$ for 2004-2010. Yin et al. (2018) have further shown that, at latitudes between 60°N and 80°N, the relationship between SCA NBP and T has transitioned from positive in the early 1980s, to negative in recent decades, reconciling the results by Keeling et al. (1996) and Schneising et al. (2014)."

EC5: Please go very carefully through all reviewer comments and ensure that everything you discussed in your response is present and clarified in your revised article!

[revised manuscript text omitted]

---

## Author Response (AR3)

**MS acp-2019-252**

Dear Editor, please note that we have updated the affiliations and acknowledgements, added labels to Figures 3 and 4 and corrected the caption of Figure 4. Apart from these changes, the manuscript text and figures are exactly the same as in the version of the MS accepted, as shown by the track changes document in attachment.

[revised manuscript text omitted]